# Making consultation meaningful: Insights from a case study of the South African mental health policy consultation process

Debra Leigh Marais[1]◐*, Michael Quayle[2,3‡], Inge Petersen[4‡]

**1** Undergraduate Research Office, Faculty of Medicine and Health Sciences, Stellenbosch University, Cape Town, South Africa, **2** Centre for Social Issues Research, Department of Psychology, University of Limerick, Limerick, Ireland, **3** Department of Psychology, School of Applied Human Sciences University of KwaZulu-Natal, Pietermaritzburg, South Africa, **4** Centre for Rural Health, College of Health Sciences, University of KwaZulu-Natal, Durban, South Africa

◐ These authors contributed equally to this work.
‡ These authors also contributed equally to this work.
* debbiem@sun.ac.za

**Data Availability Statement:** Data cannot be publicly shared because there are ethical restrictions on sharing the complete data set. The data comprises interview transcripts of qualitative,

## Abstract

### Background

It is widely recognised that mental health policies should be developed in consultation with those tasked with their implementation and the users affected by them. In the South African legislative context public participation in policymaking is assumed, with little guidance on how to conduct consultation processes, nor how to use consultation inputs in policy decisions.

### Methods

The South African Mental Health Policy Framework and Strategic Plan was adopted in 2013 after an extensive consultation process. Focussing on the 2012 provincial and national consultation summit, this case-study conducted key informant interviews and undertook documentary analysis to explore the process through which consultation inputs were–or were not–transferred to inform this policy. Between 2013 and 2016 seven interviews were conducted, and 11 documents (policy drafts and summit outputs) and transcripts of 23 audio-recorded sessions from the national summit were analysed.

### Results

Findings revealed that no substantive changes were made to the mental health policy following the consultation summits. There do not seem to have been systematic processes for facilitating and capturing knowledge inputs, or for transferring these inputs between provincial and national levels. There was also no further consultation regarding priorities identified for implementation prior to finalisation of the policy, with participants highlighting concerns about policy implementation at local levels as a result. This represents a lost opportunity for greater involvement of service users in policy development.

verbatim responses from participants. This is a small group of participants that would be potentially identifiable from their narratives and the way they have spoken about their roles and involvement in the mental health policy consultation process in South Africa, despite the use of participant codes to de-identify the data. Sharing the full data set would violate the principle of confidentiality. For researchers who meet the criteria for access to confidential data, the Biomedical Research Ethics Committee at the University of KwaZulu-Natal (South Africa) can be contacted to request access to the data (contact: Ms Anusha Marimuthu, +27312604769, BREC@ukzn.ac.za).

**Funding:** The authors received no specific funding for this work.

**Competing interests:** The authors have declared that no competing interests exist.

## Conclusions

Together with poor service-user representation, the format of the consultation process limited participant interaction and the possibility for engagement with, or uptake of, more experiential forms of knowledge. Several procedural elements were found to limit the elicitation and transference of consultation contributions for uptake into policy. Recommendations for future policy consultations include adapting the format of participatory processes to enable optimal use of participant knowledge, as well as greater service-user participation.

## Introduction

Good governance through mental health legislation and policies is recognised as a critical step in strengthening mental healthcare systems to respond to the growing burden of mental illness [1,2]. Within this context, South Africa's *Mental Health Policy Framework and Strategic Plan 2013–2020* [3] was adopted in October 2013. Developing this policy included extensive consultation, with consultation summits in eight of the nine provinces, culminating in a national consultation summit, where input was invited on the draft policy document. This study reports on the policy consultation process.

Generally, in democratic societies, the public has a right to be fully informed about the decisions that affect them and how those decisions are made [4]. In the mental health context, history has shown the importance of involving a range of stakeholders in mental health service planning and policies. With mental health policies globally calling for deinstitutionalisation and the integration of mental health into primary healthcare, increasing demands are placed on both primary healthcare workers and mental healthcare specialists [5,6]. It is thus key for effective policy implementation to involve healthcare professionals in the development of mental healthcare policies and programmes [7]. Within the context of South Africa's decentralised health system, it is equally important to ensure involvement of provincial- and district-level managers in policy development, to avoid disjuncture between national-level policy development and local-level implementation [8].

There is a corresponding shift towards people-centred services and enabling greater representation of mental healthcare users and their families in mental health policy development [9,10]. However, the extent of public involvement and incorporation of their views varies considerably. In South Africa, public participation in policymaking is legislated [11,12] and therefore expected, without necessarily problematising how such processes should be conducted, nor whether or how the inputs are used. This is especially so at national and provincial levels [13].

The political will for public participation in South Africa is not reflected in its implementation; many consultation exercises are used as endorsements for pre-determined decisions, without meaningful engagement or change [14]. This may partly be because of "time pressures, resource constraints, and capacity limitations" [15](p.11). It may also be because most policy processes are formulated at national level; this removes them from provincial and local administrative structures tasked with implementing them, as well as from communities who might participate in them [12]. This implementation gap between top-down consultation on national policies and provincial-level implementation has been well documented, both in South Africa [12,16,17] and internationally [18–21]. There is thus clear divergence between the goals of public participation and its effects in practice in the South African context, indicating a need to identify ways in which public inputs might have a meaningful impact on policy decisions.

Methods for public engagement require careful attention in order to fulfil mandates of policy effectiveness and participatory fairness. For mental health policy consultations, it is particularly relevant that processes are inclusive, representative, and fair. For example, consultation format/technique may affect opportunities for participants to speak and be heard, as well as how much their inputs may influence final decisions [22]. Also, mental healthcare users may have limited capacity to engage in policy discussions, particularly in developing countries like South Africa [10,18,23,24]. It is therefore essential to consider how consultation methods might enable or constrain interaction between participants and engagement with current policy proposals [25]. Furthermore, public inputs during policy consultation may be in the form of practical or experiential embodied knowledge [26,27]. This may be difficult to codify and capture in documented–and transferable–forms [28–30].

If mental health policy consultation is important for stronger mental health systems, then we need to better understand how to design and implement consultation processes that achieve the principles underlying policy consultation. What kinds of participatory processes might balance these multiple–often conflicting–voices and views so that all can be heard, and that simultaneously realise the value that all these views offer? How might we ensure that such processes can potentially influence or inform policy? These questions provided the impetus for this study. If policy consultation is a form of knowledge management [4], then it is essential that we design policy consultation that facilitates optimisation of available knowledge. Therefore, systematic management of knowledge should be a critical element of consultation processes.

Considering the complexity of reconciling context and knowledge inputs, clearly, policy consultation processes cannot just be 'business as usual'. Policy consultation thus far has been good at raising the profile of mental health and at assembling many voices, but not so effective at considering which sorts of participatory process formats might be optimal, nor how to use and move knowledge through these processes [4,31]. The South African mental health consultation summits provided a case study for elucidating the sorts of participatory processes described above, and the movement of knowledge through these processes. Attending to this movement of knowledge could facilitate aligning the *practices* and *principles* of consultation. Specifically, understanding how to design such processes so as to optimise the use and transfer of different kinds of knowledge may potentially inform future policy consultation efforts, to enable meaningful participation and be responsive to the knowledge contributions made.

## Context of study

South Africa's Mental Health Policy and Strategic Framework emerged from extensive policy consultation including mental health consultation summits early in 2012 in eight provinces: Eastern Cape, Free State, Gauteng, KwaZulu-Natal, Limpopo, Mpumalanga, North West, and the Western Cape. This culminated in a national mental health summit in April 2012, which assembled representatives from research and academic institutions, non-governmental organisations, the World Health Organization (WHO), mental healthcare user groups, mental healthcare professionals, and national and provincial government departments, to provide input on the draft policy [32]. Following the national summit, the task team that had been appointed by the Minister of Health to organise the national summit was reconvened to integrate inputs and finalise the policy (team members are listed in the final mental health policy) [3]. This included developing a strategic plan that identified priorities for implementation. The *Mental Health Policy Framework and Strategic Plan 2013*–2020 was promulgated in October 2013.

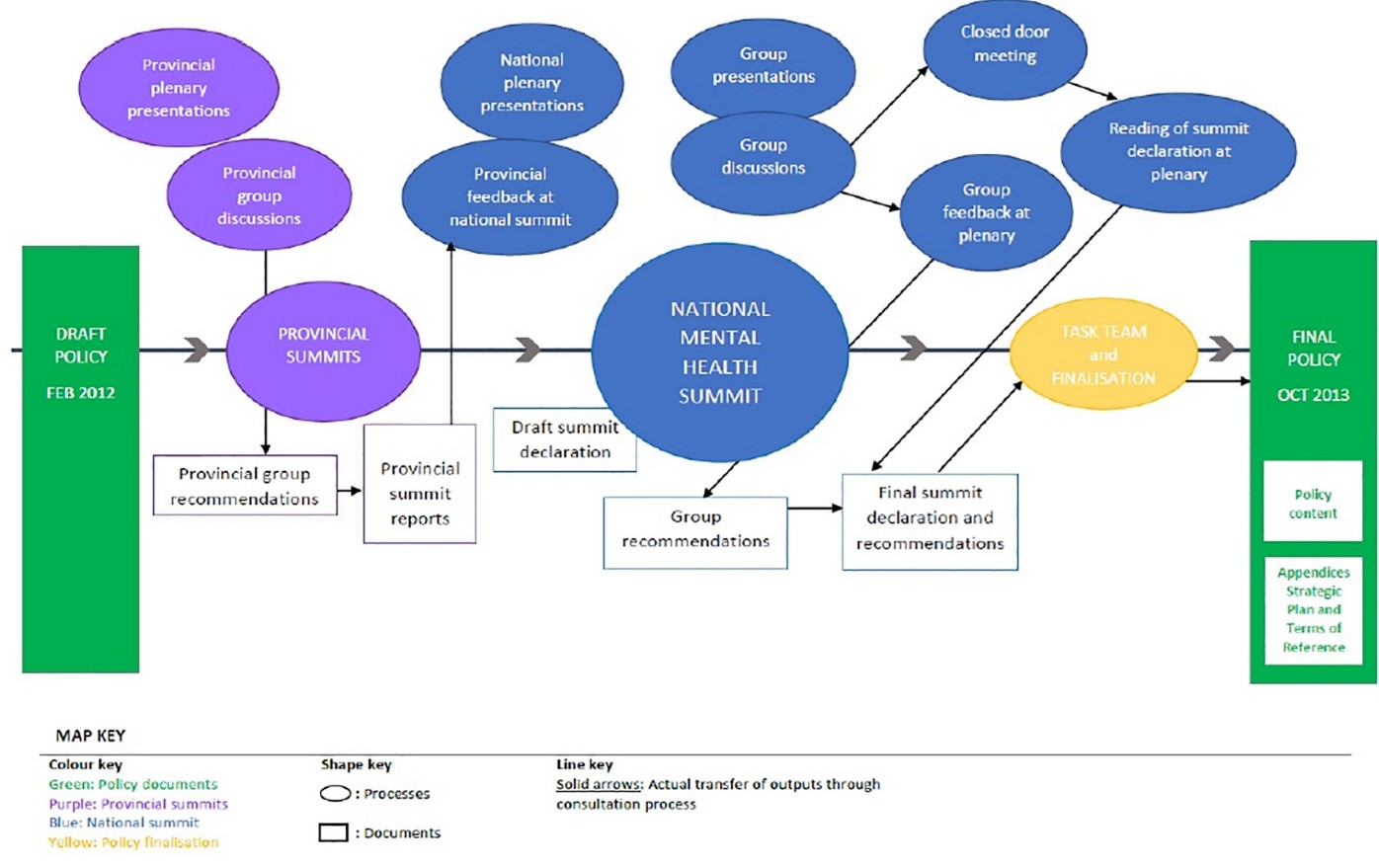

**Fig 1. Map of the mental health policy consultation process.**

A visual map of the summit policy consultation process is presented in Fig 1. This shows the chronology of the policy consultation and depicts the intersection between events or processes and their outputs. This map is based on information that could be gathered, either publicly or in the course of this study, about the policy consultation process. However, there were notable gaps in this information due to inconsistencies in both the recording and availability (and transparency) of consultation inputs across provinces [33]. The details (e.g. decisions about timing, programming, and stakeholder inclusion) are not captured here, nor is the interaction of the policy consultation process with the broader development process.

Oval shapes in Fig 1 represent events that occurred during the 2012–2013 consultation process. Rectangular shapes represent outputs from these events. In some cases, these outputs were transferred into more summarised documents or into other events; the solid arrows show this chronological movement. Colours are used to group events and outputs that were part of the same process (e.g. purple for provincial summit events, blue for national summit events).

On either side of this consultation process are the draft and final mental health policies. The starting point was the drafting of this policy, followed by the provincial consultation summits with plenary presentations and group discussion sessions. The group recommendations were an output of these group discussions, which were subsequently summarised into provincial summit reports. There was a provincial feedback session at the national summit; however, only three provinces presented their recommendations. For other provinces, it is unclear

whether or how any results of provincial-level consultations were fed back into the national process.

The national summit comprised the provincial feedback session, plenary presentations, and breakaway group sessions, with formal presentations and group discussions. Group recommendations were generated at these discussions, which were orally fed back at plenary and were also taken into a closed-door meeting where they were summarised in summit recommendations. There is no publicly accessible record of this closed-door meeting. The final summit declaration was read out at the national summit closing session.

## Methods

This study aimed to explore how the South African mental health policy consultation process informed policy, by tracing the movement of different forms of knowledge through the consultation process. Forward movement of knowledge was deemed to occur when summit inputs generated reports and recommendations to be taken forward into the final policy document. Backward movement of knowledge represented any feedback to participants regarding how their recommendations had been used, as well as further consultation on revised drafts of the policy.

This was a qualitative, instrumental case study which purposefully selected the 2012 mental health policy consultation process. Consistent with the case study imperative to include data from multiple sources [34], documents (n = 11), transcripts of audio-recordings of summit proceedings (n = 23), and key informant interviews (n = 7) comprised the data for this study. Table 1 shows the data selected for this case study, as well as the relation of each of these to the policy consultation.

### Data collection

**a) Policy documents.** In order to compare policy documents, the draft and final mental health policy documents were obtained from the National Department of Health (DoH). The

**Table 1. Data sources selected from within case.**

| Data source | Data | Relation to case context |
|---|---|---|
| **a) Policy documents** | Draft pre-summit policy (n = 1 document) | Draft policy document under review at consultation summits |
| | Final policy document (n = 1 document) | Official mental health policy (including appendices) finalised post-summit and adopted October 2013 |
| **b) Key informant interviews** | Interview transcripts (n = 7 interviews) | Retrospective process evaluation of consultation process with seven key informant participants |
| **c) Provincial and national summit documents and transcripts of audio recordings** | Provincial summit reports (where available) (n = 7 documents) | Outputs of the provincial consultation summits |
| | National mental health summit programme (n = 1 document) | Programme of events at the national mental health summit |
| | Draft summit declaration (n = 1 document) | Draft summit declaration under review at national consultation summit |
| | Transcript of audio recording of provincial summit recommendations feedback at national summit (n = 1 transcript) | Feedback of provincial summit recommendations at the national summit by provincial representatives |
| | Transcripts of ten breakaway group presentations and discussions, over two days (n = 20 transcripts) | Formal presentations and discussions that took place in each of the ten breakaway groups at the national summit |
| | Transcript of group recommendations presented at plenary (n = 1 transcript) | Feedback of breakaway group recommendations at plenary of national summit by group rapporteurs |
| | Transcript of reading out of final summit declaration (n = 1 transcript) | Adoption of finalised summit declaration, to be the formal output of the national summit |

draft pre-summit policy document was widely circulated prior to the provincial and national summits; the final policy document was available on the DoH website. The documentary analysis was conducted between October and November 2013 and focused on comparing the draft (pre-summit) policy document with the final policy document that was promulgated.

**b) Key informant interviews.** Semi-structured interviews were conducted as a retrospective process evaluation of the consultation. A generative, purposive sampling strategy was used to identify individuals who had played a substantive role in the national mental health summit. The national mental health summit programme was used to identify potential participants. Where possible, participants (from a cross-section of provinces) had participated in at least one provincial consultation summit as well as the national summit; this helped explore perspectives on the overall consultation process and follow-through of inputs/outputs. Snowball sampling was employed to identify additional participants, to include perspectives from as many of the ten breakaway group sessions as possible.

Contact was made telephonically or via e-mail in November 2013 to inform these individuals about the study and explore their willingness to participate. Interviews proceeded only after participants had provided written documentation of consent. Open-ended questions on the semi-structured interview schedule focused on the process of policy development and participants' perspectives on the consultation process, particularly on how consultation inputs informed policy.

Interviews took place in November/December 2013. All interviews were conducted by the principal author (DLM). Interviews were conducted in English, either telephonically or face-to-face, lasted one-two hours and were audio-recorded, with the participant's permission, for later transcription. Transcripts were uploaded into NVivo 10 for coding and analysis. All transcripts were de-identified and participants were assigned pseudonyms.

**c) National summit documents and audio.** Starting in May 2013, requests were sent to the national and provincial Departments of Health for documents and reports relating to the national and provincial summits that had been held in 2012.

An analysis of the transparency and variability in the ways in which the 11 provincial DoHs kept records of their consultation summits, and the extent to which they made these available, has been discussed elsewhere [33].

The data that could be obtained included reports from provincial mental health summits where available (four reports, three transcripts from oral feedback at national summit); the national mental health summit programme; and the draft and final summit declarations (national summit output). Audio-recordings from the two-day national summit were obtained from the national DoH, including the ten topic-driven breakaway group discussion sessions, which were subsequently transcribed. All participants in the breakaway group sessions were made aware that the sessions were being audio-recorded, and that these recordings may be used in subsequent reviews of the summit inputs.

The current paper focuses on the analysis of the documents and transcripts from the 2-day national summit in order to illuminate procedural aspects that may have affected the flow of knowledge inputs to inform policy. The transcription and iterative analysis of these recordings was conducted from 2014 to 2016.

## Ethical considerations

This study was approved in January 2013 by the Biomedical Research Ethics Committee (BREC) at the University of KwaZulu-Natal (ref: BE276/12), with annual renewals. It was considered to be of relatively low risk. Recruitment of participants was done on a voluntary basis; participants were informed of their right to withdraw. All reasonable steps were taken to

protect the confidentiality of participant responses. Transcripts could be linked to demographic data but not to the participant's name.

## Data analysis

**a) Preliminary phase: Policy document analysis.** The comparative analysis of the draft and final policy documents was conducted between October and November 2013. In the preliminary comparative analysis of the draft and final policy documents, content was coded thematically [35] according to the extent to which changed content in the final document represented an addition, deletion or other change from the draft policy (see S1 Table).

**b) Second phase: Interview analysis.** Second phase: Thematic framework analysis was employed to identify key themes in the interview data relating to the process of policy consultation. Framework analysis provides a means of structuring data so that themes identified in the thematic analysis phase can be summarised and compared across individual data units (i.e. the interviews) to identify commonalities and differences [36]. In conducting the various thematic analyses in this study, the Braun and Clarke guidelines [35] were used: generating an initial coding framework, identifying and refining themes within the data by sorting and categorising codes, and analysing the data within these themes in order to tell a story about the data (see S2 Table).

**c) Third phase: National summit audio transcripts.** The national summit audio-recordings were transcribed using simple transcription. The analysis focused on the transcripts from the ten breakaway group discussions. Each group had two formal presentations, followed by discussions during which participants were instructed to formulate recommendations. Each of these components was coded (as introduction, presentation, or discussion) and the percentage of the total group session spent on each component was calculated. Thematic content analysis was used for the analysis of documents and transcripts from the consultation summits. The identification of the deductive themes was guided by procedural-related aspects related to consultation processes as identified in the literature (see S3 Table). To calculate the proportion of time spent on direct engagement with policy documents, the whole session was coded for direct engagement with documents and, within this, talk was coded for whether this was engagement with the draft policy or the draft summit declaration. Calculations were based on percentages of total time. A similar process was followed for talk by chairs referring to an awareness of the need to formulate recommendations, and talk by chairs referring to the process through which recommendations needed to be formulated.

## Results

### Phase 1: Changes to the mental health policy following the consultation process

A comparison of the draft and final policy documents showed that 45 changes were made to the draft policy content following the consultation summits, while two appendices were added. Initially, it was unclear whether these changes were a direct result of discussions and requests for changes at the provincial and national summits, or a result of the work done on the policy by the Technical Advisory Committee following the consultation summits. Many changes appeared to be editorial, with little substantive effect on the policy content. There were twelve minor changes, eight deletions, and 25 additions. The majority of the changes (n = 13) were made to the *Roles and Responsibilities* section, followed by the *Areas for Action* (n = 11) and *Glossary of Terms* (n = 10) sections.

The biggest change between the draft and final policy was the addition of the eight-point *Strategic Plan*, *2013–2020* (hereafter referred to as the Strategic Plan, or implementation plan) appendix. This plan outlined the strategic actions to be implemented to effect the requirements of the mental health policy. A second substantive addition was the appendix detailing *Terms of Reference for Key Structures*. Apart from these additions, it appears that most of the changes made to the final policy would not have had a significant effect on the nature or directives of the policy.

An initial tentative conclusion drawn from this comparative analysis is that the consultation summits were simply a rubber-stamping exercise, rather than a genuine dialogue regarding the content of the mental health policy. Therefore, the focus of this study shifted to how the policy consultation process itself had unfolded. Consequently, key informant interviews were conducted to gain insight into the process and how contributions were moved between and beyond the provincial and national consultation summits, to understand why this seemed to have little direct influence on the final policy.

## Phase 2: Key informants' insights into the consultation process

Thirteen participants were invited to participate in the key informant interviews; seven agreed (six women, one man) to participate (see Table 2). Descriptive details are intentionally broad to minimise possibility of identification. Participants were based in four provinces, and their roles comprised three categories. Several attempts were made to gain participation of the conveners of the summit (DoH officials), without response. The lack of representation from the national Department of Health (DoH), as well as from all provinces, is acknowledged as a limitation.

Six main themes were identified in the thematic analysis of the interview transcripts. These were: 1) impact of national consultation summit: signalled priority; 2) impact of national consultation summit: influence on policy; 3) gaps in follow-through from provincial to national summits; 4) feedback regarding finalisation of policy following national summit; 5) opportunities for service-user involvement; and 6) perspectives on the final policy: implementation issues. The sub-themes that emerged in each of these themes are discussed further below.

## Impact of national consultation summit

Two main sub-themes emerged from the interviews regarding the impact of the national consultation summit and particularly how much it informed the final policy. Some participants emphasised the consultation summits as significantly signalling the prioritisation of mental health by national government, and as endorsing the mental health policy process by the World Health Organization (see Table 3). These were considered important ends in themselves, despite little substantive change in the policy document thereafter.

**Table 2. Participant information.**

| Pseudonym | Gender | Location | Role |
|---|---|---|---|
| Bryanna | Female | Gauteng | Service-user/representative |
| Chantal | Female | Gauteng | Service-user/representative |
| Charles | Male | Western Cape | Academic/researcher |
| Ingrid | Female | KwaZulu-Natal | Academic/researcher |
| Sameera | Female | KwaZulu-Natal | Mental healthcare practitioner |
| Sarah | Female | Western Cape | Mental healthcare practitioner |
| Zama | Female | Eastern Cape | Mental healthcare practitioner |

**Table 3. Impact of national summit sub-theme 1: Signalled priority of mental health.**

| Sub-themes | Participant responses |
|---|---|
| Signalled prioritisation of and commitment to mental health by national government/Minister of Health | There was a high level of political commitment, [and] the national Minister came, for the morning of the first day and then came back again, I think, in the afternoon of the second day (Charles). |
| | There seems to be a commitment from the national Minister of Health, because he was at the summit and he says he wants this implemented (Sameera). |
| | For the first time, the prioritisation was attached to the power of political will, in the national Minister [of Health]. And people said 'finally. We've got the Minister's ear'. So, if we can keep the minister's ear, we would love that. Because his word can make things shift (Sarah). |
| | The fact that the call came from national Department of Health says to us that, somewhere along the line, somebody realised there was a problem. We're hoping it was the national Minister of Health himself (Zama). |
| Signalled endorsement of policy and policy process by World Health Organization | The other thing that happened which was really good was that the Director of Mental Health and Substance Abuse at the WHO came to the South African national summit, which was a big thing, for a WHO person to come to a single country's policy process [and] showed the Minister that this is a really important area (Charles). |
| Raised profile of mental health | The purpose of the summits was, I think, a genuine, it was a genuine thing. The Minister did want to engage around this issue. . .I think the purpose of the summit was to raise the profile of mental health (Sarah). |

Participants had diverse perspectives regarding how much inputs at the national summit influenced the final policy (see Table 4). Some highlighted key national summit recommendations that were subsequently included (e.g. establishment of district mental health teams, in the Strategic Plan appendix). Others felt that the consultation process, as well as the availability of a draft summit resolution at the start of the summit, indicated that the process had been more of a rubber-stamping exercise than a genuine intention to engage in dialogue to review the policy. This was partially confirmed by observations that the policy document did not appear to change substantively following the consultation summits; however, the draft policy already reflected many of the issues raised at the summit.

## Gaps in follow-through from provincial to national summits

All interview participants had been involved in the provincial mental health summits, in addition to the national summit; they were asked about whether and how recommendations from provincial summits had been transferred to the national summit. The overall finding was that this transfer process was not systematic; it was either not done, was inconsistent, or there was no clear link between provincial summit outputs and national summit inputs and outputs (see Table 5).

## Feedback regarding finalisation of policy following national summit

Participants were asked about their access to information about the policy finalisation process following the consultation, including whether further consultation had occurred (see Table 6). Findings revealed that even directly involved participants were either unclear about the finalisation process (and how summit inputs had been used), or had only heard about the official

**Table 4. Impact of national summit sub-theme 2: Influence on policy.**

| Sub-themes | Participant responses |
|---|---|
| The summit was not a genuine consultation, so did not influence the policy | Through this process, we've realised that consultation doesn't always mean involvement and dialogue. Consultation really is government telling you what is their plan and then implementing that plan (Bryanna). |
| | Everyone has a different understanding of what a summit should be. For the NPO sector, we felt, it should have been a dialogue. You know, looking at unpacking all the problems and then being able to in a workshop set up, come up with strategic ways forward. But that didn't happen. You know, the summit was more, um, it was cast in stone. There was discussion, there was a lot of objection and a lot of issues were raised from the floor. But even that, couldn't always be tabled (Bryanna). |
| | And what concerned me is that when we got to the summit in the morning, we were already handed a sheet . . . basically it was the resolution of the summit. It was already printed out. This was before the summit started. So now I was really taken aback. I said now listen, if the resolution has been drawn up, then what's the point? Because my idea was that we all come in there and we deliberate issues and then you draft your resolutions and you go forward. Then I realised that this was just a kind of, rubber-stamping to say that they had done consultation. You kind of, you know, did something and then you realise you have to go back and make sure it's supposed to have been the process so you kind of, in hindsight go through it so that you can tick the boxes (Sameera). |
| The summit did not seem to result in any changes to the policy | No, I think it was, it did serve a purpose. What came out in the, both the provincial and the national summit, that reflects in the, in the draft policy that I saw. But ja. I don't know how much was changed from the draft to the, the final one (Chantal). |
| | They were fairly minor, the overall framework and the structure was pretty much the same, was pretty much intact, and, you know there [hadn't] been a lot of changes to that document as far as we could work out . . . I need to actually go back and check the extent to which the summit recommendations actually found their way into the final policy. So there's the policy document, there's the summit document, and then there's the action plan. And both the summit and the policy document fed into the action plan. But the extent to which the summit document got integrated into the policy document, I don't know. I think the substance of the summit recommendations were not that different from the policy document (Charles). |
| | Then the real policy came out and it was the same policy. The policy doesn't seem to have been amended in any way, so I can only make that comment, I can't say whether it was or wasn't (Sarah). |

(*Continued*)

**Table 4.** (Continued)

| Sub-themes | Participant responses |
|---|---|
| The way the summit was run gave participants the chance to give input and the policy did change in some key ways | So, it gave everyone the opportunity to workshop issues and then there's sort of the key issues that came out of those workshops were then summarised and put into this declaration. Then it was summarised in the backroom and then fed back . . . I don't think there was time [to do it another way]. And that was fine. I mean, I think in the end the product's really good (Ingrid). |
| | There were one or two very important issues that were clearly highlighted following the summit. And one of them relates to the establishment of district mental health teams. Such a concept was not even considered, let alone included in the last draft, but following the summit, that was one of the very important changes that I picked up in the latest draft (Zama). |
| | So, one of the recommendations we made was that in all training, of different health professionals, not just mental health professionals, it's crucial to actually focus on language, and to make sure that when, from first year, whatever the discipline the person is training in, if it's going to be health services, they should actually learn the most predominant African language spoken there . . . I think that was mentioned [in the policy document] (Zama). |

adoption of the policy 'accidentally'. Some participants felt that the identification of the eight implementation priorities included in the Strategic Plan reflected more of a service-provider than a service-user perspective.

## Opportunities for service-user involvement

There appears to have been limited service-user representation at the provincial and national summits; interview participants attributed this to several factors (see Table 7), for example, lack of the necessary (government) support (funding, accommodation, and capacity-building) that would have facilitated service-user participation. Participants felt that service-user advocates had to be proactive to involve themselves. Another factor was stigma regarding the perceived inability of service users to speak for themselves. Overall, service-user involvement was perceived as tokenistic. As a result, the policy appeared dominated by a service-provider voice, excluding some relevant issues.

## Perspectives on the final policy: Implementation issues

Regardless of participants' perceptions of the policy consultation process, their perspectives on the final policy document were unanimously positive, suggesting that a good policy can emerge from a flawed process. However, a major concern involved implementation of the policy (see Table 8); success of the final policy would only be revealed through effective implementation. Participants noted the disconnect between national policy development and provincial-level implementation, with some linking this back to the absence of systematic follow-through of provincial summit inputs to the national summit. This highlights the tension between moving from particular contexts of provincial needs and resources, to the abstract level of policy, and back again in implementation.

**Table 5. Follow-through of information from provincial to national summits.**

| Sub-themes | Participant responses |
|---|---|
| Incomplete or unsystematic transfer of information | And we felt that even though submissions were made at a provincial level, not all that information was taken through at the national summit (Bryanna). |
| | I think the main problem perhaps is the consultation that happened at provincial level, you know, perhaps not having sufficient voice at the, at the national level (Ingrid). |
| Lack of clarity regarding whether or how information was transferred | I'm not sure of the process within the Department [of Health] that led to provincial level recommendations feeding into the national process . . . I don't think they were, although that may have happened through some other forum (Charles). |
| | So basically [we had] a list of recommendations from a provincial level . . . I don't know if it was ever sent to national because we were still collating it subsequently . . . So, I'm not quite sure what the process was (Sameera). |
| Insufficient space for provincial feedback at national summit | I can't remember! I think yes, we kind of, the resolutions from the different provinces was presented there (Chantal). |
| | There was somebody in charge, in the provincial office, of collating or summarising all of those views and there was a, there was a stage when, during the national summit, there was feedback from provinces. Some provinces were not as well represented, some provinces never managed to, um, hold their provincial summit, but those who did hold it were at least able to give some feedback (Zama). |
| Lack of transparency and consistency in how information was transferred | I didn't see the feedback at the national summit. I mean, it might have been there . . . but I get a sense that it wasn't very visible (Ingrid). |
| | The provincial submissions were presented [at national] . . . but it didn't show continuity for me (Bryanna). |
| Transfer of information dependent on individual participants | There was no direct talking to between the provincial summit inputs . . . it wasn't a, kind of a, synchronised process. . . . It would have just depended on if you had a representative from your province who was at one of the [group] commissions (Sameera). |
| | What was important at the national summit was to then make sure that if you were from a particular province, and you came with that, sort of, feedback from your province, when broke away into the different sessions, it would have been important to make sure that, in your session, you carry through what your provincial, um, input would have been (Zama). |

## Phase 3: Insights into procedural issues at the national summit

Analysis of the summit documents and transcripts revealed several insights into aspects of the summit process possibly influencing the extent of summit inputs appearing in the final policy.

The national mental health summit took 16 hours over two days, structured around several components. At the end of Day 1, the ten breakaway groups convened for a one-hour session. These groups were to discuss aspects of mental health in relation to the draft policy, consistent with the themes discussed at the provincial summits. The themes of each breakaway group are shown in Table 9. On Day 2, the ten breakaway groups met for three-and-a-half hours to discuss the draft policy document and a draft declaration (the output of the summit). Thus, the breakaway groups were allocated a quarter of the total summit time. The groups produced 125 recommendations, reported back at a Day 2 plenary.

Breakaway group chairs and rapporteurs then went into a closed-door meeting with the DoH organisers, during which the 125 group recommendations were summarised into 11 recommendations for inclusion in the final summit declaration. Following the national summit,

**Table 6. Information/consultation regarding finalisation of policy document.**

| Sub-themes | Participant responses |
|---|---|
| Lack of information regarding finalisation of policy | We waited because that policy was supposed to have been launched in the media, on 10 October. That didn't materialise. Eventually we got wind that the policy was already circulated to the provincial coordinators. And it is the provincial coordinator in Cape Town where I got hold of that policy, the official policy. Even as a technical advisory committee member, I hadn't got that policy first. And that was very disturbing for me (Bryanna). |
| | It's a bit confusing it, with the launch. Because they would've launched it in the Free State, né? About . . . 3 weeks ago I think? Or a month? Then a few days before, it's cancelled. So that's where I lost of track of what's happening with this thing . . . Is this final one available on the internet, do you know? Please send it. Cos' the other day I was actually trying to find it online and there was nothing (Chantal). |
| | Ok, the final document doesn't look bad. It is fairly comprehensive. But it just would be nice if there's a documented process. You know, generally they say that, first you invite submissions on this thing, there's some kind of a formal procedure, and then you have a first draft of things, people comment, then you send it in, then you get a revision, then you get a second draft, etc. etc. So, I'm not sure, maybe that process was, maybe I'm not one of the people that was consulted. So, it may well be there. But I don't think that that process has been made transparent (Sameera). |
| Lack of consultation on implementation plan/final policy | But I think even that last part of developing the policy itself, I wasn't so much really involved in, that final thing. And maybe that should've also been, stretched to there. You know, the involvement of all the parties in the final development of the policy itself (Chantal). |
| | And then this year sometime we got a thing, draft mental health [implementation] plan. I don't know how that was arrived at. So, I'm just saying there was no, like, back and forth giving inputs etc. We got the draft plan and then subsequently I think it's now been passed so that's implemented and that's your national policy. And that's the sum total of our involvement with the national one (Sameera). |
| | I think it's nice for each province to know this, these are the people that constitute the committee [task team], these are their areas of expertise, etc. What principles guided them in terms of constituting the national task team. And you should have frames of reference, etc. For me, I would think that's the way one should go about it in terms of policy development. Then communicating exactly who's on that. So, you know, listen, that these are the experts in this respective fields. Because we know that certain people will have certain kind of, inclinations as far as certain things go. So, I think it's part of transparency when you know that these are the people that constituted that task team (Sameera). |
| | Then that committee was put together, and from that process, eight areas were flagged and now accepted for implementation. And, it's eight good things that were selected. However, there may have been one or two other things that people would have liked to have seen in there. Again, that eight-point plan was never consulted. You know, the people in that committee will highlight what is important to them. The strongest voices in a committee will hold sway. So, it's not a bad document; the priorities are some priorities. But, it's the priorities of that committee. It wasn't consulted (Sarah). |

the summit declaration was issued, containing the summit's policy recommendations (The Ekurhuleni Declaration on Mental Health, 2012).

## Time availability in breakaway group sessions

The way in which time was allocated in each of the breakaway group sessions was determined using the audio-recordings of each group session, and the proportion of talk representing each component was coded and calculated as a percentage of the total talk time. Some groups spent

**Table 7. Opportunities for service-user involvement and input.**

| Sub-themes | Participant responses |
|---|---|
| Limited service-user representation at summits | I think what it lacked was service-user involvement. For us, that was the biggest void . . . And the policy would have been very proactive and very human rights orientated if service users were given the chance (Bryanna). |
| | I had some issue with them because they put a limitation on the amount of mental health care users attending. Cos' I felt they should have had more. They should make space because ultimately, it's about, us, we with mental illness . . . Maybe also someone from a rural area, because I can't really speak of their experiences, you know, from different aspects (Chantal). |
| Service-user involvement dependent on initiative of service users/service-user organisations themselves | But through our involvement, because I knew of the summits happening in the different provinces, it was easier to inform . . . our service users in the province to say there will be a call, here's the, you know, the schedule. So, some of our NGOs had to actually contact the department and say I want to be invited to the summit . . . If we know it's happening, we take the initiative and we get involved (Bryanna). |
| | I got us on the mailing list of the Ministry. So that is the only way I know that there's this policy up for review. But that's now me, what about other service users, you know? They don't know about what there is (Chantal). |
| More support required from government for service-user participation | They didn't want to pay for the support staff, for service users. And that's lack of understanding, what does a mental health care user require to be able to participate. So, if you're flying a service user out from Cape Town, they need support staff. And we had to get into arguments with the Department, to say well, you haven't made provision for, service-user support and it was like, why do they need support? So it's a lack of understanding even from the Department side (Bryanna). |
| | With all these things, it's always very short notice, doesn't give you enough time to really prepare for it. That's always a problem. Especially when you have to review policies. . . they would tell you, the deadline is in two days, but then the document is this thick [shows with hand] so, you know, you need to go through all that. And, let's say, I had to present it [to] mental health care users. Means I now quickly need to consult with other mental health care users because I need to get their view as well. So, it makes it a bit difficult (Chantal). |
| | It would have been nice to have had a stronger mental health service-user input, but I think that reflects that nature of how service-user organisations are configured at the moment. They're not a strong advocacy lobby group; I think we should be doing more to try and support them to take on that role (Charles). |
| Service users not involved because of negative perceptions | I think even globally, people still think, you know, people with mental illness can't speak for themselves. And, even come up with resolutions themselves, you know? (Chantal). |
| | And I think the biggest barrier is the still prevalent view that if you have a mental illness, somehow you can't engage around these issues. You know, which is not true. People can and do engage. It's just that the available avenues for their engagement was not that accessible to them. Either because nobody is inviting them, or in my case, they were invited, but we didn't support their participation (Sarah). |

(*Continued*)

**Table 7.** (Continued)

| Sub-themes | Participant responses |
|---|---|
| Involvement of service users that did occur was tokenistic | And although they invited service users, a declaration was written up, without service-user involvement. And what happened in the end, a document was given to them, and said, read it. It wasn't even discussed with them. It was, here's a declaration, you go and read it. And I think that is a slap in the face (Bryanna). |
| | The people that came and gave a talk, to open it, they gave key-note presentations, in the plenary, it was Dr So-and-so from the University of XYZ, it was Professor So-and-so from the Organisation of ABC, and so it went on and then it came to the last person, and there it was just Joe Bloggs, service user. There was no organisation affiliation, he was a different animal to all the rest. So, all he needed to do there was come and stand there, doing what, representing, what was he doing? It's nice to have a service user come and tell you a story, but nobody else was telling their stories! Now he comes with his story and they say, wow, wasn't that quaint. It's not appropriate. So, there's a lot of work to be done (Sarah). |
| Policy not as representative of a service-user focus as it could have been | Service-user involvement was for us the biggest absence, the biggest void. Knowing that we had service users, even on the technical task team, it would have been . . . I think it's important to know that, you know, service users were part of that. And we've got brilliant voices in the country around service-user advocacy. And the policy would have been very proactive and very human rights orientated if service users were given the chance (Bryanna). |
| | In terms of gaps in the policy, I think it would have been nice to have had a stronger mental health service-user input (Charles). |
| | If you look at it, it's mainly about the service-provider voice. And powerful voice, always sticks out. Now service-provider voices are hugely strong. They legitimate voices. They have decades of 'this is how we do things' behind them. We're used to putting up district teams and working like this, and having HR, you know, and, knocking out the budget, and, that's the easy part . . . What hasn't been addressed is our philosophy of mental health care. You know, mental health care, is primarily been psychiatric . . . So, this policy gives us an opportunity to flip that on the head, and say, psychiatry is a strand of what needs to be delivered for people's recovery. And that's why I'm emphasising that recovery is a barrier to policy implementation, because it's a completely different thing from what we're used to (Sarah). |

most of their sessions on formal presentations and associated discussions; others allocated less time to presentations and more to group discussions and formulation of policy recommendations. The group session chairs were clearly aware of the time constraints on the group work (see Table 10); this, and the need to formulate recommendations, limited opportunities for meaningful discussions.

## Facilitation of breakaway group sessions

Each group had a chair and a rapporteur and was structured as a large meeting: contributions rotated between participants by way of turns at a microphone. This facilitated managing a great deal of input from a large number of people; however, discussions were often stilted, with little continuity between points, which limited the co-creation of new knowledge. This

**Table 8. Perspectives on the final policy: Implementation issues.**

| Sub-themes | Participant responses |
|---|---|
| Implementation and monitoring | I think, yes, a lot of the policy's quite impressive. The policy can be implemented. For me it's, after the policy, that's where we are right now and I think that's, that's the biggest issue for me. . . The problem is, the strategy for implementation. Policy has been written. But we've got to get it very clear who will monitor that implementation of that policy. You can't just email the policy and expect implementation. There's a lot of strategy and, and guidance that needs to go with it (Bryanna). |
| | You know, it's a nice policy all in all. . .I just hope with the policy, that there will be monitoring and implementation. That there would be a system in place to actually look at that. Cos it doesn't help if you develop a policy just for the sake of having a policy and it's not implemented and monitored effectively (Chantal). |
| | It's nice, everything looks nice on paper, but how are you going to effect that. And I think that's the acid test of that policy. So, it's nice to see that such a document has arrived, but it's not worth more than the paper it's written on unless it's implemented, it's changing things on the ground. So, I'm not being pessimistic. Just cautious (Sameera). |
| Implementation at provincial level | [There will be] provincial roadshows, where we meet with provincial health directors, and set out the requirements of the mental health action plan, what is expected from the provinces and really engage with them about how to do this. And I think a lot will depend on who comes to those meetings, you know, does the head of health for the province come, or do they deputise it to somebody else (Charles). |
| | It's a great policy and plan, but there does seem to be this gap between what's happening at national and what's happening at provincial. And I think the whole idea of having provincial summits leading to a national summit was great to try and bridge that gap. But I really am concerned about going forward now; how do you get your provinces to actually embrace it and dedicate resources to now being able to implement this plan beyond just these specialist teams (Ingrid). |
| | The implementation plan, is now going come to the provinces for implementation, I believe there's going to be a roadshow to introduce it. I'm not sure if national government has a plan to identify certain key things that will be funded in an extraordinary way over and above the usual allocations to province, but provinces will really have, you know, free rein, to implement those eight to ten things in the way that they see fit. Whereas if one had consulted that document with the provinces, and come to consensus around what the key issues are and what the time frames are, then you kind of can hold the provinces to what they said they would do (Sarah). |
| More detail required for implementation | But it's the how. You know, we've been saying all of this from 1994; this is what we should be doing. We need to integrate into primary health care. We need to do task shifting. I mean we've been doing it, you know. But what we need to do is actually identify the roles and functions of all the different health care personnel in the health care system in relation to mental health . . . So, you know, in terms of the structure for a mental health care plan at a district level, that's what the plan doesn't have, is who will do, what to actually implement. Now, maybe it's not supposed to be at a national level. But again, it's like, you know, we must do this and we must do that, but, how to do it at district level needs to be made clearer for the districts, I think (Ingrid). |
| | They've drafted this national mental plan. I mean, they've issued this plan but what is the implementation plan? So, it's one thing to have a document, but now what does it mean for the man [sic] on the ground? Apart from guiding us in terms of what needs to be done, it needs to tell us how it's going to get done (Sameera). |
| | In the policy, they state just one sentence which says each district must have a district mental health team. But because of how things work especially in the Eastern Cape province, if you do not sit and define what you mean by district mental health team, you may have a scenario where a psychiatrist gets employed for a district and that's your team. So, I think it would be nice to actually have a very specific statement that says, for a district mental health team, you need a minimum of, and then list, you know, what you need (Zama). |

**Table 9. Summit breakaway group themes.**

| Group no. | Breakaway group themes |
|---|---|
| 1 | Prevention and promotion |
| 2 | Research and surveillance |
| 3 | Mental health systems |
| 4 | Infrastructure and human resources |
| 5 | Mental health and other conditions |
| 6 | Mental Health Care Act |
| 7 | Child and adolescent mental health |
| 8 | Culture and mental health |
| 9 | Suicide prevention |
| 10 | Advocacy and user participation |

meeting format meant that the process in all breakaway sessions depended on the microphone's movement around the room.

Chairs' facilitation styles influenced group processes, thereby enabling or constraining opportunities for interaction (see Table 11). Facilitation style generally matched the group's microphone management strategy. Three general styles of facilitation emerged, reflecting chairs' engagement with group inputs during discussions: i) active engagement (clarifying, reframing, summarising); ii) predominantly microphone management, with little direct engagement with inputs; and iii) silence or microphone management until the second half of the session when recommendations were formulated.

There were arguably advantages/disadvantages to each facilitation style. Active engagement helped structure the discussions and increase the likelihood of capture; however, the chair could then have significant influence over what was noted and captured. Conversely, mostly silent chairs may have facilitated greater fluidity in discussions, resulting, however, in somewhat arbitrary capturing of inputs and recommendations.

## Engagement with policy documents

In general, groups seemed to engage more with the draft summit declaration than with the draft policy document (see Table 12). The mandate for the groups to generate concrete recommendations meant that much of talk was focused more on formulating recommendations than on engaging with the policy document or with one another.

As can be seen in Table 12, the total proportion of breakaway group time spent on engaging directly with the draft policy documents was just over five percent; as an average across groups, the majority of this time was spent engaging with the summit declaration.

## Formulation and capturing of recommendations

As noted, groups used different processes to develop recommendations, which was somewhat determined by how group chairs structured and facilitated the group sessions (see Table 13). The availability and quality of audio-recordings of these group sessions also varied, with four groups having very poor recordings, making inputs and process difficult to follow in the analysis.

As shown in Table 13, there was variability across groups in terms of the proportion of "formulating-recommendations" talk that was spent on comments demonstrating an *awareness* of needing to formulate recommendations, versus the proportion of this talk that was spent talking about the *process* by which recommendation should or would be formulated. Interestingly,

**Table 10. Time availability in breakaway groups.**

| Group* | Indications of awareness of time availability and limitations | | Proportion of time spent on components of group sessions | | |
|---|---|---|---|---|---|
| | Awareness of time % | Comment examples of awareness of time | Intros | Presentations | Discussions |
| 8 | 6.59% | The two papers presented are good but justice was not done to them. Time that they were presented could not afford us to comment and to critique where possible. (Speaker 6) | 17% | 26% | 57% |
| 6 | 2% | Are there any points now that if you don't make this point, the sky is gonna fall on our heads? Because otherwise we can go on and we'll miss out on the plenary. (Speaker 3) | 0 | 51% | 49% |
| 10 | 1.86% | We've got very limited time, okay . . . We have to finish this. We *have* to go for a meeting at half past, so if we can just move on. (Speaker 28) | 4% | 17% | 79% |
| 7 | 1.68% | Guys, we're going to have time problems. So I'm going to suggest if you have two points, make them briefly, so that you can give other people a fair chance. (Speaker 1) | 18% | 32% | 50% |
| 4 | 1.1% | Sorry, we really need to follow our plan. We've had our ten minutes. (Speaker 2) | 0 | 42% | 58% |
| 5 | 0.93% | I'm concerned about the time, and that we need to get through other recommendations as well. (Speaker 2) | 3% | 32% | 65% |
| 3 | 0.67% | Okay, we have to stop because we won't finish . . . But I do think we need to break now and everybody go . . . and drink five minutes of tea and come back as soon as possible. (Speaker 1) | 0 | 17% | 83% |
| 9 | 0.62% | We've got five minutes by the way . . . Now I'm in big trouble because I'm late [to closed door meeting]. (Speaker 1) | 0 | 37% | 63% |
| 2 | 0.6% | We only have 15 minutes left. Is this relevant? Is this a relevant issue? (Speaker 3) | 8% | 20% | 72% |
| 1 | 0.12% | Any other points? Everybody else is having tea, I'm just trying to let you know . . . Last point now and then we need to stop. (Speaker 9) | 4% | 31% | 65% |

* Corresponding breakaway group topics: 1. Prevention & promotion. 2. Research & surveillance. 3. Mental health systems. 4. Infrastructure & human resources. 5. Mental health & other conditions. 6. Mental Health Care Act. 7. Child & adolescent mental health. 8. Culture & mental health. 9. Suicide prevention. 10. Advocacy & user participation

exactly half of the total averaged time across all ten groups was spent on each of these two types of talk.

Groups also differed regarding the extent to which group participants could confirm or co-formulate recommendations (see Table 14). In only two groups was the group rapporteur vocally active during group discussions, clarifying inputs and acting as co-facilitator. In some groups, the rapporteur remained mostly silent during group discussions but became vocally active during formulation of recommendations. In other groups, the rapporteur was completely silent; here it seems that the chair adopted a more active role in capturing discussions and recommendations.

The number of recommendations put forward by each group varied from 5 to 27, with no apparent correlation between number of recommendations and how they were captured during group sessions. Some rapporteurs gave oral reports on the notes and recommendations. In other groups, recommendations were captured as they were being formulated, facilitating increased opportunities for participant engagement in formulating recommendations.

## Discussion

### Purpose of consultation

The point of departure for this study was the finding that the mental health policy did not change in any substantive way following the summit. This suggested that the consultation process may have been more a rubber-stamping process than a genuine dialogue with participants regarding the proposed policy. The conference-style format, common in South Africa, has been used previously to endorse "emerging legislation" [37](p.190). It seems this consultation

**Table 11. Facilitation of breakaway groups.**

| Group[*] | Microphone management references | General procedural comments by Chair | Chair engagement with inputs |
|---|---|---|---|
| | **Examples of microphone mgt. comments** | **Examples of general procedural comments** | **Description** |
| 6 | Sorry, just before I go to you, can I go to the gentleman with the blue shirt? Because your hand's been up. (Speaker 1) | It's very important that if you make a comment that you use the microphone, otherwise your comments might not be recorded and I think it's important that we have an accurate recording of the proceedings here. (Speaker 1) | Active engagement (clarifying, reframing, summarising) |
| 10 | So, I'm gonna go . . . firstly, I saw that hand first. And then I saw a hand over there as well. . . (Speaker 28) | Are there any other burning issues before we divide up into groups? Oh, we've got a very burning person here. And another burning person there–is it very burning? (Speaker 28) | Active engagement (clarifying, reframing, summarising) |
| 7 | I'm going to jump to my two senior colleagues here because I know they talk a lot and they have a lot to offer, so let me move on to this side. (Speaker 1) | Now, if you have other specific thing, without making long speeches, just say it so the rapporteur can capture it. (Speaker 1) | Mostly managing microphone |
| 3 | There's a hand at the back. (Speaker 1) I even stood up. (Speaker 22) Sorry, and I still ignore you . . . Do you want to come pick up the mic? (Speaker 1) | I don't want us to get into comments. Remember we were not even supposed to have a debate or a speaker tonight so we need to finish. (Speaker 1) | Active engagement (clarifying, reframing, summarising) |
| 1 | Nobody wants the mic? (Speaker 9) (name) wants to talk; (name) wants the power again. (Speaker 1) The power of the mic. (Speaker 9) | Okay, so really need to try and keep people's focus on one issue. (Speaker 1) | Active engagement (clarifying, reframing, summarising) |
| 4 | So, we have one here, here, and then I know you were next . . . I have an extra mic for those who wants to speak. (Speaker 2) | Just to reassure you, I'm going to give a very brief summary to (Name of organiser). This presentation still takes place in plenary. But I just want to make sure that the very brief summary I give to them now meets what we have discussed (Speaker 2) | Mostly managing microphone until end of session; active in formulating recommendations |
| 2 | Just pass the microphones around . . . Do you want to repeat that for the mic? (Speaker 1) | I'm getting worried. Can you assume your responsibilities, Chair? You know, others are not going to be given the opportunity to interact in this commission. (Speaker 76) | Mostly managing microphone until end of session; active in formulating recommendations |
| 9 | Now, we need to apparently record things here, so I'm going to have to move around a little bit. (Speaker 1) | I'm going to run quickly to my meeting with (Name of organiser). If you two would like to quickly just put your heads together, so that (Name of rapporteur) has the correct thing to feed back over there (Speaker 1) | Silent until second session; active in formulating recommendations |
| 8 | Right, I see a hand at the back. (Speaker 1) | I just want to make a plea, let's not make speeches. If you are given a chance to comment, if you make a speech, it gets boring. There are people who have been designated to give speeches. (Speaker 1) | Active engagement (clarifying, reframing, summarising) |
| 5 | Sorry, there's someone who wants to speak over there. (Speaker 2) | I think we need to maybe focus less on the difficulties and more about where do we think it's reasonable to get, and how are we going to get there. (Speaker 2) | Silent until second session; active in formulating recommendations |

* Corresponding breakaway group topics: 1. Prevention & promotion. 2. Research & surveillance. 3. Mental health systems. 4. Infrastructure & human resources. 5. Mental health & other conditions. 6. Mental Health Care Act. 7. Child & adolescent mental health. 8. Culture & mental health. 9. Suicide prevention. 10. Advocacy & user participation

process was more an information exchange than an exchange of power between citizens and government through mutual decision-making; this may have amounted to tokenistic participation [38]. Thus, if outcomes were predetermined, then how knowledge transfer was effected through the consultation process was inconsequential. However, to optimise the value of the consultation, management of information through the process should at least have been systematic.

Several findings suggested that the consultation outcome was at least partially predetermined (e.g. the presence of a pre-drafted summit declaration). Previous local and international research has shown that consultation can leave participants feeling disempowered due to being co-opted into processes with predetermined outcomes [12–14,39]. The finding that the 125 group recommendations were reduced to the eleven recommendations on the summit

**Table 12. Engagement with draft documents in breakaway groups.**

| Group* | | Engagement with policy | | | Engagement with summit declaration | |
|---|---|---|---|---|---|---|
| | % of total document engagement talk | Policy refs (% of total) | Description of extent of engagement with draft policy | Summit decl. refs (% of total) | Description of extent of engagement with draft summit declaration |
| 8 | 18.05% | 40% | Direct detailed engagement with during discussions | 60% | Direct detailed engagement with during discussions |
| 9 | 7.13% | 0% | No reference to or direct engagement with during discussions | 100% | Direct detailed engagement with during discussions |
| 3 | 6.03% | 58% | Direct engagement | 42% | Direct engagement with |
| 1 | 4.86% | 42% | Direct detailed engagement; framed presentation and discussion around this | 58% | Referred to briefly at end |
| 2 | 2.22% | 17% | Referred to in instructions only | 83% | Referred to in instructions only |
| 10 | 1.29% | 100% | Referred to in instructions only | 0% | No reference to or direct engagement during discussions |
| 7 | 0.35% | 0% | No reference to or direct engagement with during discussions | 100% | Referred to in instructions only |
| 6 | 0.34% | 0% | No reference to or direct engagement with during discussions | 100% | Referred to in instructions only |
| 4 | 0% | 0% | Instructions unknown; no direct engagement during discussions | 0% | Instructions unknown; no direct engagement during discussions |
| 5 | 0% | 0% | Instructions unknown; no direct engagement during discussions | 0% | Instructions unknown; no direct engagement during discussions |
| % of total break-away group time | 5.03% | 32.12% | | 67.88% | |

* Corresponding breakaway group topics: 1. Prevention & promotion. 2. Research & surveillance. 3. Mental health systems. 4. Infrastructure & human resources. 5. Mental health & other conditions. 6. Mental Health Care Act. 7. Child & adolescent mental health. 8. Culture & mental health. 9. Suicide prevention. 10. Advocacy & user participation

declaration behind closed doors–with no audio or documented records of this meeting–also suggests a top-down process, whereby government retains control over both decisions and decision-making. This confirms findings of studies of health policymaking and consultation, in South Africa [7,8], and elsewhere [20,21,40].

This study also adds to South African research showing that public participation in practice tends to be more tokenistic and less empowering than promised in legislative mandates [13]. This may be a capacity and feasibility issue [14] rather than a disingenuous government agenda. However, tokenistic policy consultations have particularly negative ramifications in the context of mental health, as this may further marginalise already vulnerable groups [27]. Arguably, the consulted public should not be misled that their inputs will directly influence final decisions [40,41]; nevertheless, some evidence suggests that the national consultation summit had positive consequences for mental health in South Africa.

### Follow-through and feedback

The findings suggest that both forward movement (provincial to national summit; breakaway groups to the technical task team responsible for finalising the policy) and backward movement of inputs through the consultation process was lacking. Backward movement was lacking in terms of feedback to consultation participants regarding use of their inputs in finalising the policy and identifying implementation priorities.

Firstly, the transfer of provincial summit recommendations for national summit consideration was inconsistent, depending on individuals from the provincial summits being present in the group discussions and ensuring that these inputs were heard. This is particularly

**Table 13. Formulation of recommendations in breakaway groups.**

| | Awareness of need to formulate recommendations | | Process through which recommendations would be or were being formulated | |
|---|---|---|---|---|
| Group* | Awareness of need to formulate recommend-ations | Comments demonstrating awareness of need to formulate recommendations | Process for formulating refs (% of total) | Description of process of getting to recommendations |
| 8 | 56% | Obviously, we will not have a shopping list that would be a hundred demands. We will need to come up with a very limited number of issues. (Speaker 1) | 44% | Direct engagement with wording of draft documents and suggested changes, which formed basis for recommendations |
| 3 | 19% | I first want a solution. Nobody's going to talk unless they talk about what's the target. Alright? (Speaker 1) | 81% | Participants put forward recommendations on paper; Chair took majority and directed discussion on formulating recommendations |
| 10 | 25% | Just to remind you and those that joined us later, tomorrow's very outcomes based for us to give input into this policy. How are we going to make advocacy a reality? (Speaker 1) | 75% | Broke into four small groups to discuss recommendations proposed by Chair; small group discussions not audio recorded |
| 7 | 50% | I understand that this session, what we need to do is to try to add to the points that you've made with some specific targets that the Department of Health can adopt. (Speaker 9) | 50% | Formulated during second half of session by rapporteur, with input from participants |
| 9 | 18% | (Name of organiser) just said to us yesterday that please, when we come with those proposals, they must be reasonable, they must be achievable; it mustn't be completely bizarre. (Speaker 1) | 82% | Formulated during second half of session by Chair, with inputs from one or two participants |
| 6 | 35% | I think that what would be very important for the summit would be to be able to move away from the summit with some key proposals that came from this group in terms of achievable and realistic objectives that could be implemented. (Speaker 3) | 65% | Formulation of recommendations began at start of discussions and continued throughout |
| 5 | 63% | Going forward the rest of this time, we actually need to come up with targets in terms of what we want to achieve in terms of mental health management of chronic diseases. (Speaker 2) | 37% | Chair proposed five major recommendation categories; directed discussion to formulate specific recommendations around these |
| 4 | 34% | Please ask yourself, are your comments taking us forward into resolution to possible ideas? (Speaker 2) | 66% | Pieces of paper collected ad hoc |
| 2 | 100% | Just keeping in mind that we have to have something concrete to feedback at the plenary . . . declarations rather than a wish list because I think it just won't happen. (Speaker 3) | 0% | Formulation of recommendations at end |
| 1 | 100% | So we should definitely add something to that. Can you formulate something then? (Speaker 1) | 0% | No explicit formulation of recommendations |
| Total % of talk about recommendations | 50% | | 50% | |

* Corresponding breakaway group topics: 1. Prevention & promotion. 2. Research & surveillance. 3. Mental health systems. 4. Infrastructure & human resources. 5. Mental health & other conditions. 6. Mental Health Care Act. 7. Child & adolescent mental health. 8. Culture & mental health. 9. Suicide prevention. 10. Advocacy & user participation

significant as the national summit intended to inform implementation priorities, a provincial DoH responsibility. Secondly, as noted, the 125 group recommendations were 'converted' to eleven summit declaration recommendations in an unrecorded closed-door meeting. More broadly, there was apparently no systematic process for transferring inputs beyond the national summit, meaning the consultation process outputs have little potential to inform policy.

**Table 14. Capturing of recommendations in breakaway groups.**

| Group* | Role of rapporteur | Format of inscription | Number of group recommend. | % of total number |
|---|---|---|---|---|
| 4 | Feedback on notes made at various points in process; active in formulation of recommendations | Captured onto PowerPoint slides before and during formulation; shown on screen during formulation of recommendations | 27 | 20% |
| 10 | Active during discussions; individual rapporteurs from small groups reported back | Oral report back of presentations and comments on these; oral report back of recommendations from small groups by small group rapporteurs | 21 | 15% |
| 7 | Silent during discussion; active during formulation of recommendations | Oral report back by rapporteur | 17 | 13% |
| 5 | Silent | Unknown; no report back | 16 | 12% |
| 6 | Active during discussion, clarifying, and capturing | Captured and projected onto screen during discussion and formulation of recommendations | 15 (only 4 in plenary) | 11% |
| 1 | Silent; no checking back in | Unknown; no report back | 12 | 9% |
| 2 | Unknown | Captured onto PowerPoint slides; shown on screen during formulation | 9 | 7% |
| 8 | Active during engagement with documents and formulation of recommendations | Oral report back by rapporteur | 9 | 7% |
| 3 | Silent | Written down by Chair/rapporteur during formulation | 5 | 4% |
| 9 | Silent until oral report back of notes made during discussions | Oral report back on discussions by rapporteur; captured by Chair on board at front of room during formulation of recommendations while rapporteur typed | 5 | 4% |

* Corresponding breakaway group topics: 1. Prevention & promotion. 2. Research & surveillance. 3. Mental health systems. 4. Infrastructure & human resources. 5. Mental health & other conditions. 6. Mental Health Care Act. 7. Child & adolescent mental health. 8. Culture & mental health. 9. Suicide prevention. 10. Advocacy & user participation

As noted, there was no feedback to participants regarding how inputs were used to inform policy, nor any further consultation regarding how priorities for implementation were identified. There was also apparently no systematic process for informing participants that the final policy had been adopted, despite this being expected practice [41–43].

In addition, inconsistencies in provincial summit record/report format, how they were managed, and lack of public accessibility [33] makes it difficult to see how public inputs are (or are not) incorporated into policy decisions. This highlights the disparity between legally mandated public participation and its practice in South Africa. There is a need to identify mechanisms through which public inputs might have a meaningful impact on policy decisions.

## Service-user involvement

The poor representation of mental healthcare service users at the national consultation summit confirms other studies demonstrating limited service-user participation in policy development in low- and middle-income countries [10,44,45], and in South Africa in particular [23,46]. This study, as in other studies [47,48], showed that service-user involvement in the consultation summit was limited and somewhat tokenistic. This risks perpetuating existing negative perceptions and stigma regarding service users [9].

In addition, the findings indicate that certain forms of input were more acceptable than others, and that certain groups of participants (e.g. service users) were only invited to contribute in prescribed forms (e.g. reading out the final declaration at the end). Furthermore, the breakaway group sessions were framed with formal expert presentations, potentially communicating the perception that certain individuals (e.g. the presenters) had rights to make legitimate claims, while other participants did not. Thus, experiential knowledge was silenced

during the policy consultation processes, further sidelining service users or lay participants [47].

## Procedural issues

As noted, consultation inputs should be linked with policy decisions, demonstrating responsiveness to participants' recommendations [25,49]. This study showed that engagement and transfer between these knowledge forms at the mental health consultation summits was not optimal. This may be why the mental health policy did not change substantively following the consultation, despite a great deal of input.

The study findings demonstrate that the chairs of the breakaway sessions had to integrate not just multiple knowledge inputs, but also different knowledge forms (verbal, written). They dealt with this in different ways, with varying consequences for the optimisation and transfer of knowledge inputs. The findings also indicated that opportunities for knowledge contributions by participants at the national consultation summit were limited, mainly due to time restrictions. Similarly, within the group sessions, formal presentations occupied considerable time. Such time pressures reduce the effectiveness of policy consultation [50], also hindering development of recommendations [51]. Consultations aim to draw on participants' knowledge, therefore time and space must be provided for discussion, before considering the capture of inputs as recommendations. Meaningful consultation should allow for negotiating, framing, re-framing, debating, disputing, and using different forms and sources of knowledge [31].

The breakaway group organisation also influenced the movement of knowledge through the process. The national summit process mirrored the tendency in policy consultation towards conventional practices, structured like formal academic conferences, with chairs and rapporteurs managing small group sessions. Given the large number of participants, this format did facilitate the process and achieve the summit's objective of producing a report with a feasible number of recommendations [49,52]. However, if knowledge is created in interaction, then this group session format also limited opportunities for the creation of new knowledge, as well as the nature of such knowledge. This meant that many other potentially valuable voices and knowledge inputs were not provided with sufficient 'airtime', and thus could not reach the recommendations.

If consultation is genuinely intended to elicit participants' views on policy proposals under consideration, then the interactive nature of such processes should be optimised. This means attending to exclusion mechanisms that limit participation, including the setting, chosen methods of communication, and speaking time granted [53]. The findings of this study support similar studies that highlight the importance of attending to process in consultation spaces [25,47], as well as to creative tools for eliciting and capturing knowledge inputs [52,54]. However, this study shows that facilitation needs to extend beyond process towards being able to integrate multiple knowledges [31,55].

## Policy-implementation gap

Adequate consultation processes are essential for policy implementation [56], especially in terms of how inputs move between provincial and national consultation events and back again [57]. One consequence of a decentralised health system is that the development of policies at national level removes them from the provincial and local administrative structures which implement them, as well as from participating communities [8,12,58]. Lack of consultation around implementation priorities may negatively affect policy implementation at provincial and district levels [8,12]. Consultation plays a valuable role in policy development; this study supports research highlighting this role [59,60].

## Limitations

The gaps in access to data and information regarding the overall consultation process is likely to have limited what could be inferred from the findings. The lack of transparency about the changes made to the policy following the summit and particularly the process followed by the Technical Advisory Committee tasked with finalising the policy meant that it was not possible to determine how many of the changes made were as a result of outputs from the summit, or other factors. The small number of participants and lack of representation from a number of the provinces as well as the national Department of Health meant that the perspectives on the consultation process may have been somewhat skewed. It may be that a more representative sample might have provided a more diverse range of perspectives, particularly given the variability across provinces. A representative from national DoH might have provided valuable insight into many of the 'gaps' identified in the flow and nature of information from the consultation summits. Regarding the transcript analysis, although in most cases there were audible audio-recordings for the ten breakaway group sessions, there were instances in which a particular session's recording was either not available or the audibility of a particular recording was poor. This limits the conclusions that could be drawn across all groups with respect to the procedural issues discussed in this paper.

## Conclusion and recommendations

The findings of this study have provided insights into how knowledge moves through a policy consultation process, highlighting several gaps in consultation at the intersection of knowledge and policy. Policy consultation represents (among other things) an exercise in knowledge management. Considerable research has studied factors that contribute to the effectiveness of policy consultation; however, current literature offers little guidance in terms of micro-level processes that might optimise the use of consultation inputs towards enhancing policy and policy implementation. This study provides several insights to fill this gap.

This study shows clearly that participatory processes should be designed with greater attention to knowledge management (from eliciting to sense-making to capturing to transferring). Also, this study argues for more systematic transfer of inputs through the consultation process, both for forward movement of inputs (consultation summits to policymakers), as well as for 'backward' movement of information (feedback from policymakers to consultation participants/public). Without a systematic process for moving knowledge through and beyond the consultation space, consultation is unlikely to influence policy, even indirectly.

The lack of explicit influence of the South African mental health policy consultation process on the policy itself has several implications. The predetermined nature of some of the summit outcomes suggests that the consultation may have been intended more to secure endorsement of the draft policy than to change it, consistent with previous research showing the primarily tokenistic culture of public participation in South Africa.

This consultation was apparently also a lost opportunity for realising the value of summit participants' knowledge contributions for enhancing mental health policy and implementation. Such conventional forms of policy consultation may not provide optimal spaces for authentic engagement or follow-through of inputs. One alternative participatory process is an open-space design format, allowing for greater interaction between large numbers of participants, as well as engagement with present policy proposals. However, this requires trained facilitators, and the ability to understand and capture the participants' detailed inputs, in order to optimise the value of consultation for policy development.

Similarly, this consultation process was a missed opportunity for enabling greater service-user participation and its potential influence on policy; this remains an elusive ideal. Service

users were poorly represented at the consultation summits and the process limited their knowledge contributions to tokenistic inputs. Increased efforts must be made to structure future mental health consultation processes to facilitate service-user participation.

The findings show clearly the disconnect between national and provincial authorities regarding policy consultation and formulation. There was no systematic process during the mental health consultation summits for linking provincial-level inputs to the mental health policy with national-level inputs. This is problematic because South African provincial health departments must implement policies developed by the national DoH. Therefore, the transfer of knowledge from sites of implementation to sites of consultation, and back to implementation sites though policy, requires greater attention in order to strengthen the mental health system.

Consultation practices should enable multiple types/forms of knowledge to potentially inform policy–particularly implementation priorities–in order to reconcile the gap between national-level decision-making processes and what happens on the ground. This particularly affects those providing healthcare and those who live with and care for people with mental illness. Thus, future consultation events should be held at more local levels, with the explicit objective of gathering inputs on how the policy might work on the ground.

Furthermore, mechanisms should be established through which ongoing consultation and feedback can occur, particularly with respect to identifying priorities for implementation that are locally relevant and feasible. There have been significant challenges with implementation of the South African mental health policy. It is therefore critical that future policy consultation processes attend more carefully to the management of knowledge inputs to ensure that the voices of those tasked with implementation–from individual to provincial level–are reflected in policy priorities.

## Supporting information

**S1 Table. Codes for analysis of draft and final policy documents.**
(DOCX)

**S2 Table. Codes for analysis of interview data.**
(DOCX)

**S3 Table. Codes for analysis of procedural issues at national summit.**
(DOCX)

## Acknowledgments

The authors gratefully acknowledge the assistance of Mrs Viv O'Neill in editing this paper.

## Author Contributions

**Conceptualization:** Debra Leigh Marais, Michael Quayle.

**Data curation:** Debra Leigh Marais.

**Formal analysis:** Debra Leigh Marais.

**Methodology:** Debra Leigh Marais, Michael Quayle, Inge Petersen.

**Supervision:** Michael Quayle, Inge Petersen.

**Writing – original draft:** Debra Leigh Marais.

**Writing – review & editing:** Debra Leigh Marais, Michael Quayle, Inge Petersen.

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
