## [Decision Letter · Decision Letter 0]

30 Oct 2019

PONE-D-19-25777

Making consultation meaningful: Insights from the South African mental health policy consultation process

PLOS ONE

Dear Dr Marais,

Thank you for submitting your manuscript to PLOS ONE. After careful consideration, we feel that it has merit but does not fully meet PLOS ONE’s publication criteria as it currently stands. Therefore, we invite you to submit a revised version of the manuscript that addresses the points raised during the review process.

Please carefully address the comment of both reviewers, by playing special attention to the major aspects to be improved prompted by reviewer 2 and that concern more methodological aspects. 

We would appreciate receiving your revised manuscript by Dec 14 2019 11:59PM. To enhance the reproducibility of your results, we recommend that if applicable you deposit your laboratory protocols in protocols.io, where a protocol can be assigned its own identifier (DOI) such that it can be cited independently in the future. For instructions see: http://journals.plos.org/plosone/s/submission-guidelines#loc-laboratory-protocols

We look forward to receiving your revised manuscript.

Kind regards,

Sara Rubinelli

Academic Editor

PLOS ONE

Journal Requirements:

Reviewers' comments:

Reviewer's Responses to Questions

**Comments to the Author**

1. Is the manuscript technically sound, and do the data support the conclusions?

Reviewer #1: Yes

Reviewer #2: Yes

2. Has the statistical analysis been performed appropriately and rigorously? 

Reviewer #1: I Don't Know

Reviewer #2: N/A

3. Have the authors made all data underlying the findings in their manuscript fully available?

Reviewer #1: Yes

Reviewer #2: No

4. Is the manuscript presented in an intelligible fashion and written in standard English?

Reviewer #1: Yes

Reviewer #2: Yes

5. Review Comments to the Author

Reviewer #1: Thank you for letting me review this article, which focuses on the process of Mental health policy development and implementation in South Africa. To explore this process, the article authors have appropriately opted for a qualitative design, and the article is overall well-written. I find this article interesting and I only have some minor comments and questions.

1. Abstract and Methods: the abstract does not include any methods description. But this needs to be included. The abstract should preferably include for example the number and type of documents reviewed. The number of transcribed audio-recordings analysed, and the sampling and number of key informant interviews. As well as the period of data collection and analysis. This type of information that can give the reader a better insight into the research process such as the amount of data analysed and how thorough the process was carried out. This form of summary information could also be given in the methods section, e.g. on line 156, page 7.

2. Line 186, page 8. The data collection period included the review of documents, the transcription of audio-recordings and key informant interviews? The key informant interview data collection started November 2013 (emails sent out) and ended in December 2013 (line 192, page 9). When did the review of documents and transcription and analysis of audio-recordings start and end?

3. Methods: line 163-165, page 7. About the audio-recordings. It would be interesting to know a little more about consent in relation to the audio-recorded data. Were conference participants aware of that they were being recorded and that their discussions might be used for research, audit or similar?

4. Methods: line 192, page 9. Who undertook the key informant interviews?

5. Methods: line 219-225, page 10. It would be interesting to know here, in relation to the audio-recordings from the two-day summit, how the time calculations (that is, the descriptive percentages presented in Table 8,10 and 11) were made using the transcribed audio-recording information. This is briefly mentioned elsewhere in the article. However, this could be made clearer and placed so that the information is easy to find for the reader of the article.

6. Methods: Table 9, page 26. It is stated in the article that the key informant interviews were anonymised. Looking at Table 9 on page 26, there is a name mentioned in the Table; is it correct it the case that the audio-recordings from the two-day summit also were anonymised? In other words, this is a name given by the article authors and not the real name of a summit participant? This should be made clearer.

7. Data analysis, key informants. 1) Table 1 on page 8, outlines the role of the study participants (the key informant interviewees). However, in Table 2-7 names of interviewees are being used. It would, I believe and if possible, be more relevant and interesting to use the key informants’ roles and gender (could e.g. be marked with Role and then “F” and “M” for Female/Male) rather than their names in relation to their quotes.

8. Data analysis, transcribed audio-recordings from the two-day summit, line 392, page 23. it would be helpful if the authors could add to this section of the article that the information concerning “Time availability in breakaway group sessions” derive from the audio-recordings.

9. Discussion/limitations. Each method used, that is, the review of documents, the audio-tape transcript analysis, and the analysis of the key informant interviews should preferably be discussed in terms of their potential limitations. Whilst this is mentioned in the article, it would be good with a brief paragraph discussing the main limitations, encompassing all methods used, in the discussion part of the article too. Did the authors’ feel that they had been able to analyse all data sufficiently? If not, what would have been good to pay further attention to? What about the gender distribution in the key informant interviews, in what way might this have impacted on data analysed? And so forth.

10. Proof reading. There are some small mistakes here and there in the article. For example, 1) there appear to be a piece of cited text on line 61-62, page 4, but no page number to the reference that the quote is from. The authors should go through the article to ensure that quotes etc. are referenced correctly. 2) The reference list needs a little more attention. For example, reference 2 include information such as “copyright” and “All rights reserved”; reference 4 and 33 do not state which Government the reports concern or in what country the reports were published, and reference 41 has a “*” in its article title, which doesn’t seem to fit. 3) The text in Table 3 on page 13 and 14, has gaps in it (after the word “So” in Table 3 on page 13, and after the word “then” in Table 3 on page 14).

Reviewer #2: Thank you for the opportunity to review this paper. This case study reports on the underlying mechanisms of public consultation in health policy, and conducts a documentary analysis and key informant interviews to explore the process through which consultation inputs were/ were not transferred to inform this policy. Overall, the study was well written, however, given the length and amount of details in this case study; I have included here some points for consideration, to help with the overall clarity of the paper.

Title

1. Include within the title that it is a case study

Abstract

2. It would improve clarity if the abstract is divided into background, methods, results & conclusion

Methods

3. Details on the task team that finalized the report, or who commissioned this team would be helpful.

4. Reorganize methods subsections for more clarity and consistency into the 3 phases:

a. Policy documents

b. Key Informant Interviews:

i. Participant Recruitment (Also moving lines 185-187 “The national mental health summit programme was

used to identify potential participants. Contact was made telephonically or via e-mail in November 2013 to inform these individuals about the study and explore their willingness to participate.” within this section

ii. Data collection

c. Provincial and national summit documents and audio

5. Data analysis in the third phase is a bit confusing. The focus seems to be on the ten breakaway group discussions from the national summit, however there is no data analysis mentioned for the provincial mental summit reports and transcripts.

6. Identify interviews as “semi-structured interviews”, since an interview schedule was used.

7. From your results, it seemed that you also performed some triangulation between different methods (“this was partially confirmed by observations that the policy document did not appear to change substantively”) – it would be helpful to mention at the end of your data analysis section whether there was any effort in comparing or triangulating between different methods, as this is often done in case studies.

Results

8. The number of participants in the key informant interviews and characteristics (Table 1: Participant information) should be included in the results instead of methods.

9. Phase 2 includes a variety of sub-sections (i.e. Impact of national consultation summit, Gaps in follow-through from provincial to national summits..). An introduction to how these were derived would be helpful – were these based on the interview schedule? Are these all themes (in which case, language within each of these “themes” should be consistent).

10. On line 258, “Two main themes” are mentioned. I would move (see Table 2) to the next line, as those 3 sub-themes arise from Theme 1. Additionally, in labeling Tables 2 and 3, it would be helpful to have consistent language as in the results section (i.e. Table 2: Sub-themes from Theme 1: Purpose of national consultation summit; Table 3: Sub-themes from Theme 2: How the national consultation summit informed policy”)

11. The sub-themes within “gaps in follow-through from provincial to national summits” (Table 4) overlap with one another (i.e. Lack of information about whether and how provincial recommendations transferred to national summit + Provincial feedback at national summit was not consistent or visible enough). Perhaps these are different, but if there is a way to use consistent language throughout the sub-themes, it would help with clarity. Siince all these sub-themes were discussing the transfer of information from provincial to national summits, the sub-themes could be reworded to remove redundancy. (e.g. Incomplete or non-systematic transfer of information, Minimal space for feedback, Dependent on individual participants, Unsure about extent of information transfer).

12. Include “Phase 3” with the Insights into procedural issues at the national summit sub-title.

13. To improve readability of the results in phase 3, you can include 1 or 2 interesting examples of “awareness of time” comments, and just leave in the quantitative % in Table 8. Additionally, the breakaway group topics could be included within the results text of Phase 3.

14. Table 11 seems to be missing text in the column header (Awareness of need to formulate)

15. For tables 11 and 12, it would be helpful to report on the number of "direct engagement" vs "recommendations formulated by chair" and “silent rapporteurs” vs active rapporteurs” instead of a list of the 10 breakout groups. Since the purpose of your case study is to identify the underlying methods or mechanisms of the consultation process (and not a comparison between the 10 groups) you could do some further grouping in these 2 tables to identify the different strategies or processes used.

Discussion

16. The concepts of “forward movement” and “backward movement” of inputs is introduced in the discussion – these would be helpful concepts to introduce in the introduction or methods

17. A breadth of methodological limitations are reported throughout the paper (i.e. inconsistencies in recording and availability, transparency; lack of representation from the DoH/ provinces, poor quality audio recording) – it would be helpful to summarize these in the discussion to identify any limitations with answering your research questions.

Minor grammatical

18. Line 90: change “available knowledges” to available knowledge

6. PLOS authors have the option to publish the peer review history of their article (what does this mean?). If published, this will include your full peer review and any attached files.

Reviewer #1: Yes: Dr Karin Johansson Blight, RGN

Reviewer #2: No

---

## [Author Response · Author response to Decision Letter 0]

28 Nov 2019

Academic Editors’ Comments

1 Please ensure that your manuscript meets PLOS ONE's style requirements, including those for file naming. 

The style guide and file naming requirements have been carefully attended to and the manuscript and files amended accordingly. 

Comments from Reviewer 1

1 Abstract and Methods: the abstract does not include any methods description. But this needs to be included. The abstract should preferably include for example the number and type of documents reviewed. The number of transcribed audio-recordings analysed, and the sampling and number of key informant interviews. As well as the period of data collection and analysis. This type of information that can give the reader a better insight into the research process such as the amount of data analysed and how thorough the process was carried out. This form of summary information could also be given in the methods section, e.g. on line 156, page 7 

This detail has been included in the abstract and in the methods section. 

2 Line 186, page 8. The data collection period included the review of documents, the transcription of audio-recordings and key informant interviews? The key informant interview data collection started November 2013 (emails sent out) and ended in December 2013 (line 192, page 9). When did the review of documents and transcription and analysis of audio-recordings start and end? 

Detail regarding when each component of the data collection and analysis has been added to the methods section (subsections a-c). 

3 Methods: line 163-165, page 7. About the audio-recordings. It would be interesting to know a little more about consent in relation to the audio-recorded data. Were conference participants aware of that they were being recorded and that their discussions might be used for research, audit or similar? 

The issue of consent for the recording of the summit breakaway sessions has been included in the methods section. 

4 Methods: line 192, page 9. Who undertook the key informant interviews? 

The principal author conducted the interviews. This detail has been included in the methods section. 

5 Methods: line 219-225, page 10. It would be interesting to know here, in relation to the audio-recordings from the two-day summit, how the time calculations (that is, the descriptive percentages presented in Table 8,10 and 11) were made using the transcribed audio-recording information. This is briefly mentioned elsewhere in the article. However, this could be made clearer and placed so that the information is easy to find for the reader. 

Detail regarding time calculations has been added to the methods section. 

6 Methods: Table 9, page 26. It is stated in the article that the key informant interviews were anonymised. Looking at Table 9 on page 26, there is a name mentioned in the Table; is it correct it the case that the audio-recordings from the two-day summit also were anonymised? In other words, this is a name given by the article authors and not the real name of a summit participant? This should be made clearer. 

Thank you for pointing this out. The name that appeared in the table (now Table 11) has been removed.

7 Data analysis, key informants. 1) Table 1 on page 8, outlines the role of the study participants (the key informant interviewees). However, in Table 2-7 names of interviewees are being used. It would, I believe and if possible, be more relevant and interesting to use the key informants’ roles and gender (could e.g. be marked with Role and then “F” and “M” for Female/Male) rather than their names in relation to their quotes. 

Thank you for this point, which helped us to make the information included in the table providing participant information (now Table 2) clearer. We have chosen to add the pseudonyms to the table so that this can serve as a reference point for the tables that follow, in which we have kept the pseudonyms to refer to each participant. 

8 Data analysis, transcribed audio-recordings from the two-day summit, line 392, page 23. it would be helpful if the authors could add to this section of the article that the information concerning “Time availability in breakaway group sessions” derive from the audio-recordings. 

This detail has been added to the results section. 

9 Discussion/limitations. Each method used, that is, the review of documents, the audio-tape transcript analysis, and the analysis of the key informant interviews should preferably be discussed in terms of their potential limitations. Whilst this is mentioned in the article, it would be good with a brief paragraph discussing the main limitations, encompassing all methods used, in the discussion part of the article too. Did the authors’ feel that they had been able to analyse all data sufficiently? If not, what would have been good to pay further attention to? What about the gender distribution in the key informant interviews, in what way might this have impacted on data analysed? And so forth. 

A summary of limitations has been added to the discussion section. 

10 Proof reading. There are some small mistakes here and there in the article. For example, 1) there appear to be a piece of cited text on line 61-62, page 4, but no page number to the reference that the quote is from. The authors should go through the article to ensure that quotes etc. are referenced correctly. 2) The reference list needs a little more attention. For example, reference 2 include information such as “copyright” and “All rights reserved”; reference 4 and 33 do not state which Government the reports concern or in what country the reports were published, and reference 41 has a “*” in its article title, which doesn’t seem to fit. 3) The text in Table 3 on page 13 and 14, has gaps in it (after the word “So” in Table 3 on page 13, and after the word “then” in Table 3 on page 14). 

Thank you for pointing out these errors. These have been corrected. (Note: Table 3 is now Table 4) 

Comments from Reviewer 2

1 Title: Include within the title that it is a case study

The title has been changed accordingly.

2 Abstract: It would improve clarity if the abstract is divided into background, methods, results & conclusion. 

The abstract has been divided into these sections. 

3 Methods: Details on the task team that finalized the report, or who commissioned this team would be helpful. 

This detail has been included in the Context of study section. 

4 Methods: Reorganize methods subsections for more clarity and consistency into the 3 phases:

a. Policy documents

b. Key Informant Interviews:

i. Participant Recruitment (Also moving lines 185-187 “The national mental health summit programme was used to identify potential participants. Contact was made telephonically or via e-mail in November 2013 to inform these individuals about the study and explore their willingness to participate.” within this section

ii. Data collection

c. Provincial and national summit documents and audio. 

Thank you for this recommendation which has helped to make the sub-sections and chronology of the methods clearer. The headings have been added and amended accordingly. 

5 Methods: Data analysis in the third phase is a bit confusing. The focus seems to be on the ten breakaway group discussions from the national summit, however there is no data analysis mentioned for the provincial mental summit reports and transcripts. Clarity regarding the provincial reports has been added to the methods section (sub-section c). 

The heading of the relevant section in the analysis section (sub-section c – Third phase…) has been amended to make the focus on the national transcripts clearer. 

6 Methods: Identify interviews as “semi-structured interviews”, since an interview schedule was used. 

This detail has been added to the methods section (sub-section b). 

7 Methods: From your results, it seemed that you also performed some triangulation between different methods (“this was partially confirmed by observations that the policy document did not appear to change substantively”) – it would be helpful to mention at the end of your data analysis section whether there was any effort in comparing or triangulating between different methods, as this is often done in case studies. 

Triangulation was conducted across the results from the different phases of analysis. However, this was done as a means of showing areas of convergence and divergence with respect to questions relating to a theoretical framework that was used as a lens to make sense of the movement of different kinds of knowledge through the process. This theoretical paper is in process. We believe that the detail on triangulation is thus beyond the scope of the current paper. 

8 Results: The number of participants in the key informant interviews and characteristics (Table 1: Participant information) should be included in the results instead of methods. Participant information has been moved to the results section. 

9 Results: Phase 2 includes a variety of sub-sections (i.e. Impact of national consultation summit, Gaps in follow-through from provincial to national summits..). An introduction to how these were derived would be helpful – were these based on the interview schedule? Are these all themes (in which case, language within each of these “themes” should be consistent). This clarification has been added to the results section. 

10 Results: On line 258, “Two main themes” are mentioned. I would move (see Table 2) to the next line, as those 3 sub-themes arise from Theme 1. Additionally, in labeling Tables 2 and 3, it would be helpful to have consistent language as in the results section (i.e. Table 2: Sub-themes from Theme 1: Purpose of national consultation summit; Table 3: Sub-themes from Theme 2: How the national consultation summit informed policy”). 

These amendments have been made to the text and to the tables (now Table 3 and Table 4). 

11 Results: The sub-themes within “gaps in follow-through from provincial to national summits” (Table 4) overlap with one another (i.e. Lack of information about whether and how provincial recommendations transferred to national summit + Provincial feedback at national summit was not consistent or visible enough). Perhaps these are different, but if there is a way to use consistent language throughout the sub-themes, it would help with clarity. Since all these sub-themes were discussing the transfer of information from provincial to national summits, the sub-themes could be reworded to remove redundancy. (e.g. Incomplete or non-systematic transfer of information, Minimal space for feedback, Dependent on individual participants, Unsure about extent of information transfer). 

The sub-theme labels in the table (now Table 5) have been amended to make the content to which each sub-theme refers clearer and more distinguishable. 

12 Results: Include “Phase 3” with the Insights into procedural issues at the national summit sub-title. 

The sub-heading has been changed accordingly. 

13 Results: To improve readability of the results in phase 3, you can include 1 or 2 interesting examples of “awareness of time” comments, and just leave in the quantitative % in Table 8. Additionally, the breakaway group topics could be included within the results text of Phase 3. Thank you for this suggestion. We have decided to leave the table as is, with examples of quotes from each group corresponding to the percentages, as we feel that the quotes give context to the quantitative data. 

We have added in a table (Table 9) showing the themes per breakaway group, which will hopefully provide a more user-friendly reference for the reader. 

14 Results: Table 11 seems to be missing text in the column header (Awareness of need to formulate). 

The column heading in the table (now Table 13) has been corrected. 

15 Results: For tables 11 and 12, it would be helpful to report on the number of "direct engagement" vs "recommendations formulated by chair" and “silent rapporteurs” vs active rapporteurs” instead of a list of the 10 breakout groups. Since the purpose of your case study is to identify the underlying methods or mechanisms of the consultation process (and not a comparison between the 10 groups) you could do some further grouping in these 2 tables to identify the different strategies or processes used. 

Thank you for this suggestion. For consistency we have chosen to keep the structure of the tables (now Tables 12 and 13) the same. The aforementioned triangulation and theoretical analysis will draw to some extent on the group detail, so we would prefer to keep this detail included. However, we have included totals in the tables and discussed these in the text, so that the overall proportion of time spent across all groups is clearer. 

16 Discussion: The concepts of “forward movement” and “backward movement” of inputs is introduced in the discussion – these would be helpful concepts to introduce in the introduction or methods. 

These concepts have been included in the methods section. 

17 Discussion: A breadth of methodological limitations are reported throughout the paper (i.e. inconsistencies in recording and availability, transparency; lack of representation from the DoH/ provinces, poor quality audio recording) – it would be helpful to summarize these in the discussion to identify any limitations with answering your research questions. 

A summary of limitations has been added to the discussion section. 

18 Minor grammatical: Line 90: change “available knowledges” to available knowledge.

Edit has been made.

---

## [Decision Letter · Decision Letter 1]

13 Jan 2020

Making consultation meaningful: Insights from a case study of the South African mental health policy consultation process

PONE-D-19-25777R1

Dear Dr. Marais,

We are pleased to inform you that your manuscript has been judged scientifically suitable for publication and will be formally accepted for publication once it complies with all outstanding technical requirements.

With kind regards,

Sara Rubinelli

Academic Editor

PLOS ONE

Additional Editor Comments (optional):

Reviewers' comments:

Reviewer's Responses to Questions

**Comments to the Author**

1. If the authors have adequately addressed your comments raised in a previous round of review and you feel that this manuscript is now acceptable for publication, you may indicate that here to bypass the “Comments to the Author” section, enter your conflict of interest statement in the “Confidential to Editor” section, and submit your "Accept" recommendation.

Reviewer #1: All comments have been addressed

Reviewer #2: All comments have been addressed

2. Is the manuscript technically sound, and do the data support the conclusions?

Reviewer #1: Yes

Reviewer #2: (No Response)

3. Has the statistical analysis been performed appropriately and rigorously? 

Reviewer #1: Yes

Reviewer #2: (No Response)

4. Have the authors made all data underlying the findings in their manuscript fully available?

Reviewer #1: Yes

Reviewer #2: (No Response)

5. Is the manuscript presented in an intelligible fashion and written in standard English?

Reviewer #1: Yes

Reviewer #2: (No Response)

6. Review Comments to the Author

Reviewer #1: Thank you for this revised article manuscript, I am satisfied with the changes made. Whilst the denominator for the percentages calculations, that is the "total talk time", do not seem to be mentioned in relation to the descriptive percentages in Table 12 and 13, in my opinion- on the basis of how the manuscript is written and the overall high quality in the process description- I a) trust that the authors have provided the correct percentage numbers in these tables, and b) understand that the percentages are used as a way to communicate the qualitative analysis to the readers of the manuscript in a more transparent way. For these reasons I do not think further amendments of Table 12 and 13 are necessary.

Finally, the manuscript does not entail specific (book) references to the qualitative theories used to guide this study (such as grounded theory or other). However, it appear that this has been acknowledged in the peer-review process of the manuscript, meaning the authors are aware of this and they appear to have planned to include theory in a separate paper. In addition to this, there are also several references to published peer-reviewed research made in the article manuscript, which focuses on relevant research methodology used. For these reasons, I am satisfied with the way that the theoretical framework for the research undertaken has been discussed. Overall I believe this is a well-written and interesting article.

Reviewer #2: Thank you for addressing all my comments - these edits help improve the clarity and readability of your paper.

All the best!

7. PLOS authors have the option to publish the peer review history of their article (what does this mean?). If published, this will include your full peer review and any attached files.

Reviewer #1: Yes: Dr Karin Johansson Blight, RGN

Reviewer #2: Yes: Lydia Sequeira

---

## [Editor Report · Acceptance letter]

15 Jan 2020

PONE-D-19-25777R1 

Making consultation meaningful: Insights from a case study of the South African mental health policy consultation process 

Dear Dr. Marais:

I am pleased to inform you that your manuscript has been deemed suitable for publication in PLOS ONE. Congratulations! Your manuscript is now with our production department. 

With kind regards,

on behalf of

Dr. Sara Rubinelli 

Academic Editor

PLOS ONE